

# Measurement report: Ion clusters as indicator for local new particle formation

Santeri Tuovinen[1], Janne Lampilahti[1], Veli-Matti Kerminen[1] and Markku Kulmala[1]

[1]Institute for Atmospheric and Earth System Research, University of Helsinki, Helsinki, 00014, Finland

## Abstract

Atmospheric aerosol particles have considerable impact on climate, both directly by scattering radiation and indirectly by acting as cloud condensation nuclei. A major fraction of global aerosol number is formed in atmospheric new particle formation (NPF). These atmospheric particles consist of both neutral particles and charged atmospheric ions, and atmospheric ion number concentrations have been observed to indicate NPF. In this work, atmospheric ion concentrations were studied with the aim of finding the best suited size range of ions to characterize local NPF. Both negative and positive ion number size distributions measured by Neutral cluster and Air Ion Spectrometer (NAIS) at the SMEAR II measurement station in Hyytiälä, Finland were used. Ion sizes between 1.6 and 3 nm were considered. We found that the negative ions between 2.0 and 2.3 nm are well suited for representing NPF. In addition, the influence of transport on the observed 2.0-2.3 nm ion concentrations is considerably smaller than for larger ions. Therefore, we recommend the negative ions with diameters 2.0-2.3 nm as the best choice for characterization of local NPF.

## 1 Introduction

Atmospheric aerosol particles affect climate on local, regional and global scales (Boucher et al., 2013; Rosenfeld et al., 2014; Quaas et al., 2022; IPCC, 2022). These particles scatter radiation, impacting Earth's radiative balance (Bellouin et al., 2005; Yu et al., 2006). In addition, particles with diameters larger than about 50-100 nm are able to act as cloud condensation nuclei (CCN) (Komppula et al., 2005; Anttila et al., 2010; Bougiatioti et al., 2020). CCN are a necessity for cloud droplet formation, and CCN number and properties influence cloud properties such as cloud irradiance (Rosenfeld et al., 2014; Fan et al., 2016). A large fraction of global aerosol number is produced through atmospheric new particle formation (NPF) (Spracklen et al., 2010; Gordon et al., 2017).

NPF is a phenomenon, during which new atmospheric aerosol particles form due to gas-to-particle conversion (Kulmala et al., 2001; Kerminen et al., 2018). NPF events are regularly observed all over the globe, from boreal forests to urban megacities (Dal Maso et al., 2007; Dada et al., 2017; Kerminen et al., 2018; Chu et al., 2019; Bousiotis et al., 2021; Brean et al., 2023). The particle production through NPF by a certain location depends on its environmental conditions. For example, low levels of particle pollution and sufficient abundance of potential precursor vapors such as sulfuric acid, bases and oxidized organic compounds tend to favor NPF (Paasonen et al., 2010; Kulmala et al., 2013a; Schobesberger et al., 2013; Dada et al., 2017; Kerminen et al., 2018; Lehtipalo et al., 2018; Yan et al., 2021). To properly evaluate the strength of the local particle



production, or the impact of the local conditions on NPF, the influence of particles originating from outside the area of interest should be minimized.

40 NPF has been observed to occur regularly at the SMEAR II measurement station in Hyytiälä, southern Finland, both as daytime regional NPF events, during which NPF takes place over a large region (Dal Maso et al., 2005; Nieminen et al., 2014; Dada et al., 2017), and as local evening and nighttime clustering events (Mazon et al., 2016). Over 20% of the days in Hyytiälä are classified as NPF event days (Dada et al., 2018), however to some extent NPF is expected to occur even on those 45 days, which are not classified as events based on the traditional classification schemes (Kulmala et al., 2022). At SMEAR II measurement site, NPF has been estimated to increase CCN number concentrations by more than 70% (Sihto et al., 2011).

Neutral cluster and Air Ion Spectrometer (NAIS) is able to measure number size distributions of both total aerosol particles (~2 – 42 nm) and negative and positive atmospheric ions (0.8-42 nm) 50 (Manninen et al., 2009b; Mirme and Mirme, 2013; Manninen et al., 2016). Atmospheric ions can be divided into small cluster ions (< ~1.6 nm), intermediate ions (> ~1.6 nm, < ~7.4 nm), and large ions (> ~7.4 nm) based on their size and physical properties (Tammet, 1995; Hõrrak et al., 2000; Hirsikko et al., 2011). Background concentrations of small cluster ions are present almost all the time (Hõrrak et al., 2000; Hirsikko et al., 2011; Kulmala et al., 2013a). These ions are formed 55 mainly through ionization of neutral clusters and particles (Nagaraja et al., 2003; Bazilevskaya et al., 2008; Laakso et al., 2004; Harrison and Tammet, 2008). Activation of small particles for growth, which can lead to NPF, has been estimated to occur between about 1.5 to 2 nm (Kulmala et al., 2013b). Ions larger than about 2 nm are produced by charging of neutral particles, or by ion induced nucleation followed by the growth of these ions (Hõrrak et al., 1998, 2000; Hirsikko et al., 60 2011). Atmospheric intermediate ions have been found to be a strong indicator of NPF events (Tammet et al., 2014; Leino et al., 2016). However, ions in this size range may have considerable variability in where they originate from, and might therefore be better suited for representing NPF on regional than local scales. Therefore, the ion size range for representing NPF on a local scale needs to be reconsidered.

65 In this study, we aim to narrow down the diameter range used to represent NPF to minimize the effects of transported ions on the observations. We study both negative and positive ions with diameters between 1.6 and 3 nm, and evaluate their potential for use to represent local clustering and NPF. The potential contribution of transport on the ion concentrations is discussed. Finally, a recommendation for the best ion size range to characterize local NPF with minimal influence from 70 transported ions is given.

## 2 Methods

We used ion number size distribution data from SMEAR II (Station for Measuring Forest Ecosystem–Atmosphere Relations II) measurement station (Hari and Kulmala, 2005). SMEAR II station is located in Hyytiälä, southern Finland (61°51´N, 24°17´E, 180 m), and the site is 75 surrounded by relatively homogeneous Scots pine forest. For more details on the site and measurements, see e.g., Nieminen et al. (2014).



The ion number size distributions were measured with a NAIS (Neutral cluster and Air Ion Spectrometer) (Mirme and Mirme, 2013; Manninen et al., 2016). The NAIS is able to measure both air ions (0.8–42 nm) and total particles (~2–42 nm) by the use of a corona charger. Due to the presence of charger ions in diameters up to 1.8 nm in the total particle size distributions measured by the instrument (Manninen et al., 2009b), we restricted our analysis to atmospheric ions in this study. The data were inverted using the v14-lrnd inverter (Wagner et al., 2016). The time resolution of the data was two minutes.

The ion size number distribution data used was from years 2016 to 2020. These data were used from all the available days, and no distinction was made based on whether the days had been classified as NPF days or not. Four different ion size bins, which were based on the used inversion method, were considered in our analysis (Table 1). The ion concentrations were converted to absolute number concentrations, and outliers were removed based on 1% and 99% quantiles. However, we would like to note that the effect of this procedure on our results was found to be minor.

**Table 1:** The four different size bins, which were used in the analysis. The data was measured by the Neutral cluster and Air Ion Spectrometer (NAIS) and the bins are based on the data inversion used.

| Geometric mean diameter (nm) | Limits (nm) |
| --- | --- |
| 1.87 | $1.73 \leq d_{ion} < 2.01$ |
| 2.16 | $2.01 \leq d_{ion} < 2.32$ |
| 2.49 | $2.32 \leq d_{ion} < 2.68$ |
| 2.88 | $2.68 \leq d_{ion} < 3.10$ |

## 3 Results and discussion

We investigated atmospheric ion concentrations for different diameters to determine the most suitable size, or size range, for representing local new particle formation (NPF). Ions in the sizes of a few nm have been previously used to investigate NPF (see e.g., Kulmala et al., 2013b). In this work, we narrowed the investigated diameters to between 1.6 and 3 nm. These limits were chosen based on our motivations in this study: first, we wanted the source area of the ion concentrations to be as small as possible. Thus, ions larger than 3 nm were not considered as they could originate from relatively large distances. Second, we wanted to observe clusters that were already growing to larger sizes, i.e., clusters initiating a measurable NPF. Therefore, the smallest clusters below 1.6 nm were not included in our analysis because such clusters might not lead to the formation of larger particles.

### 3.1 Diurnal cycles of ion concentrations

First, we investigated the diurnal cycles of ion concentrations in different size bins between 1.6 and 3 nm to find which sizes could be used to represent NPF. For a size bin to be suitable for this purpose, the sensitivity for clustering should be as high as possible, while the influence of transport on the concentrations should be minimized. To compare the different size bins, we determined diurnal profiles of the ion concentrations in different size bins (see Table 1) based on median, 25%,



and 75% quantile values for 30 minute time windows. These are presented in Figure 2. Figures 5 and 3 include data only from March-May and September-November, correspondingly. During March-May NPF events are regularly observed at the SMEAR II station while during September-November they are less common (Nieminen et al., 2014; Dada et al., 2018).

Most curves for ion concentrations in Figure 2 and 5 have peaks during midday, indicating the presence of daytime NPF, which is visible in all size bins. However, the difference of the peak compared to the background level is relatively small in $d_{bin} \approx 1.87$ nm. For example, for negative ions corresponding to the 75% quantile line, the daytime peak is approximately 9 cm$^{-3}$ while the lowest value in the same percentile is around 8 cm$^{-3}$. For $d_{bin} \approx 2.16$ nm, $d_{bin} \approx 2.49$ nm, and $d_{bin} \approx$ 2.88 nm, the daytime peaks are relatively similar in strength compared to the background concentrations. For $d_{bin} \approx 2.16$ nm and $d_{bin} \approx 2.49$ nm, the daytime peak in the 75% quantile line is around 60% higher than the background level while for $d_{bin} \approx 2.88$ nm the difference is slightly higher. In addition, evening peaks after 18:00 are seen at least in the higher quantiles of the negative ion mode. These evening peaks are the strongest in $d_{bin} \approx 1.87$ nm and $d_{bin} \approx 2.16$ nm. This suggests that evening clustering, which has been observed to take place at the measurement site (Mazon et al., 2016; Rose et al., 2018), is the strongest in these size bins. In Figure 3, possible evening clustering is also observed at these two size bins, however at $d_{bin} \approx 2.16$ nm the peaks are weak, indicating that although some clusters might form, few of them grow above 2 nm during the autumn season. This is likely due to low concentrations of organic precursor vapors, preventing clusters from growing larger, and it also illustrates the unreliability of sub-2 nm ions in representing particle production.

As the concentration curves of both negative and positive ions at $d_{bin} \approx 1.87$ nm in Figure 2 show relatively little variation compared to the other sizes, it appears that values in this size bin are strongly influenced by the constantly present background ions. In addition, the daytime peak, which corresponds to occurrence of daytime NPF, is weaker than the evening peak. Thus, it can be concluded that the connection of concentrations of ions in $d_{bin} \approx 1.87$ nm with NPF is uncertain. Because of these reasons, and based on previous studies (Kulmala et al., 2013b), data from $d_{bin} \approx$ 1.87 nm should not be used as an indicator for NPF.

Figures 1, 2, and 3 also show that the background concentration of positive ions is higher compared to that of negative ions, and that the relative change in ion concentrations during NPF is higher for negative ions. We postulate that the influence of constant background concentrations is larger for positive ions due to their larger mobility diameters compared to negative ions (Hõrrak et al., 2000; Harrison and Aplin, 2007), extending the background to larger diameters. Thus, negative ions are preferred over positive ions to represent local cluster production and NPF.

## 3.2 Transport of ions and the impact on ion footprint

As the main motivation of this study is to characterize local NPF, it is critical to consider the effect of transport on concentrations of ions of different diameters. Figure 4 shows how far is the distance ions with different diameters can originate from assuming some ion growth rate (GR) and wind speed. Initial size of 2 nm has been assumed based on previous studies (see e.g., Kulmala et al., 2013b).



If a wind speed of 3 m/s and GR of 2 nm/h, which is close to the average particle GR in Hyytiälä (Manninen et al., 2009a), are assumed, then observed ions in $d_{bin} \approx 2.88$ nm can originate from a distance of ~5 km (Figure 4). With the same conditions, ions in $d_{bin} \approx 2.49$ nm can have been transported from ~1.5 to ~3 km away, while ions in $d_{bin} \approx 2.16$ nm mostly originate from within a 1 km distance. With the assumed wind speed and GR, $d_{bin} \approx 2.16$ nm is the only one of these three size bins that includes ions from within a 500 m distance. However, if wind speed is 1 m/s, most of the ions in all the investigated size bins originate from within a 1 km distance, and most ions in $d_{bin} \approx 2.16$ nm can be assumed to be from within 500 meters. Potential ion footprint area is sensitive to both ion GR and wind speed. Figure 4 shows a rough estimate, and for a more accurate estimation of ion footprint area other factors such as surface roughness and canopy height would need to be considered. For our purposes in this study, a rough estimate is sufficient.

It is clear that to minimize the effect of ion transport on the atmospheric ion concentrations, while still having a reliable indicator for NPF, the ion diameter should be as close as possible to 2 nm. Therefore, based on the discussion here and in Sect 3.1 negative ion concentrations in $d_{bin} \approx 2.16$ nm appear to be the best candidate to represent local NPF. In addition, if the ion concentrations are used alongside variables derived based on eddy covariance measurements (average footprint between 500 m and 1 km) such as $CO_2$ concentrations, the footprint area of these ions is similar.

## 3.3 Impact of data amount on ion diurnal cycle

Based on the discussion in the previous section, the negative ion concentrations from $d_{bin} \approx 2.16$ nm appear to be a good indicator for local NPF, both in terms of statistical behavior and the footprint area. However, using data only from one size bin could potentially increase the influence of statistical noise, especially if data are sparse, and thus make it more difficult to make accurate observations of particle formation. Figure Error: Reference source not found shows the median diurnal curves for negative ion concentrations in $d_{bin} \approx 2.16$ nm and between the diameters 2.01–2.50 nm and 2.05–2.68 nm. The latter are based on the nearest neighbor interpolation and take into account the concentrations in both $d_{bin} \approx 2.16$ nm and $d_{bin} \approx 2.49$ nm. Both curves with all data and 50%, 10% and 1% of it are included.

Figure Error: Reference source not found illustrates two important things: first, including the concentrations only from $d_{bin} \approx 2.16$ nm or from both $d_{bin} \approx 2.16$ nm and $d_{bin} \approx 2.49$ nm has a minor effect on the averaged behavior of the negative ion concentration. Secondly, reducing the amount of data does not seem to result in a more considerable amount of noise if only data from one size bin is used compared to if data from two size bins is used. Thus, we argue that the negative ion concentrations in $d_{bin} \approx 2.16$ nm, which corresponds to a diameter range 2.01–2.32 nm, best, and with most certainty, indicate local cluster formation.

## 4 Atmospheric relevance and applicability

Our results show that negative ions with diameters 2.0-2.3 nm are the best suited for representation of local NPF. However, it is important to note that while the negative 2.0-2.3 nm ion concentrations do show the strength of NPF within a limited source area, NPF within the area does not occur in isolation. Air masses from outside the footprint area transport larger pre-existing particles and precursor chemical compounds, influencing both the rate at which growing clusters are scavenged



by larger particles and the rate that they are growing to larger sizes. For example, in Hyytiälä, air masses arriving from northwest direction have been shown to favor NPF due to these air masses having a low surface area of pre-existing particles (Dal Maso et al., 2007; Dada et al., 2017). In addition to precursor compounds emitted within the area, transported precursor compounds could also affect the number of new clusters.

Next, example cases for application of our results are discussed. First, if we want to investigate cluster or particle production in different environments, such as in a boreal forest and a wetland, the negative ion concentrations in the environments can be used with some assumptions. If the average condensation sink (CS) and ion growth (GR) rate are similar in these environments, the negative 2.0-2.3 nm ion concentrations can be used to represent the relative contribution to total particle production by these different types of environments (see Kulmala et al., 2023). This information can then for example be used to estimate the contribution of different environments to e.g., total aerosol radiative forcing. Second, we can study the impacts of different factors on cluster production: for example, if two otherwise very similar environments have considerably different averaged ion concentrations, the impacts of other factors such as CS or transport of chemical compounds from outside the ion source area on ion concentrations and local particle production can be evaluated.

In addition, the negative 2.0-2.3 nm ions originate mainly within a source area, which is similar in size to the footprint area of tower-based eddy covariance measurements. Kulmala et al. (2020) recently developed a concept of CarbonSink+, which accounts for multiple boreal forest climate-biosphere feedbacks, including atmospheric particles and NPF. Our results can therefore be applied to represent particle production within a similar area as $CO_2$ flux, and other fluxes, to study their combined climate impacts (see Kulmala et al., 2023).

While we have only used data from Hyytiälä in this study, we argue that our results are also applicable to other environments. This is because the dynamics of tropospheric air ions can be assumed to be relatively similar, and thus sub-2 nm ions are not suitable for representing particle production. In addition, although the average contribution of transported ions on negative 2.0-2.3 nm ion concentrations varies on environmental basis, this impact is as small as it can be for ion concentrations measured by NAIS in this size range, while still gaining meaningful information on NPF.

## Conclusions

In this work, we determined the best ion size to represent local NPF, with minimized impact of transported ions on the observations. To fulfill our aim, we studied ion concentrations in four different size bins between 1.6 and 3 nm measured by Neutral cluster and Air Ion Spectrometer (NAIS) at the SMEAR II measurement station in Hyytiälä, southern Finland. We found that the negative ions in the size bin with geometric mean diameter $d_{bin} \approx 2.16$ nm (2.0-2.3 nm) are the best choice to represent local NPF. Firstly, the ion concentrations in this size bin have considerably less influence of transport compared to the larger ions. With the average wind speed and GR in Hyytiälä, they can be assumed to mostly originate from within a distance of 1 km. Secondly, the statistical difference between periods when clusters are formed and when they are not formed is sufficiently high. Therefore, these ion concentrations can, with a good confidence, be considered to represent NPF within a relatively small source area both in Hyytiälä and other locations. Our results can be



applied to investigate NPF occurring within a close proximity to and observation site. In addition,
230   they can be used in investigations of complex climate-biosphere interactions, and to assess the
combined climate effects of particle production and other factors such as carbon sink.

## Data availability

The ion number concentrations used in this study are available at
https://doi.org/10.5281/zenodo.8059335 (Tuovinen et al., 2023).

## Author contributions

ST conducted the data analysis and wrote the paper. JL was responsible for the ion measurements.
VMK and MK designed the study. All authors contributed to discussion of the results and provided
input for the paper.

## Acknowledgments

240   We acknowledge the following projects: ACCC Flagship funded by the Academy of Finland grant
number 337549 (UH) and 337552 (FMI), Academy professorship funded by the Academy of
Finland  (grant no. 302958), Academy of Finland projects no. 1325656, 311932, 334792, 316114,
325647, 325681, 347782, "Quantifying carbon sink, CarbonSink+ and their interaction with air
quality" INAR project funded by Jane and Aatos Erkko Foundation, "Gigacity" project funded by
245   Wihuri foundation, European Research Council (ERC) project ATM-GTP Contract No. 742206, and
European Union via Non-CO2 Forcers and their Climate, Weather, Air Quality and Health Impacts
(FOCI). University of Helsinki support via ACTRIS-HY is acknowledged. University of Helsinki
Doctoral Programme in Atmospheric Sciences is acknowledged. Support of the technical and
scientific staff in Hyytiälä are acknowledged.

## Competing interests

250

At least one of the (co-)authors is a member of the editorial board of Atmospheric Chemistry and
Physics. Authors have no other conflicts of interest to declare.





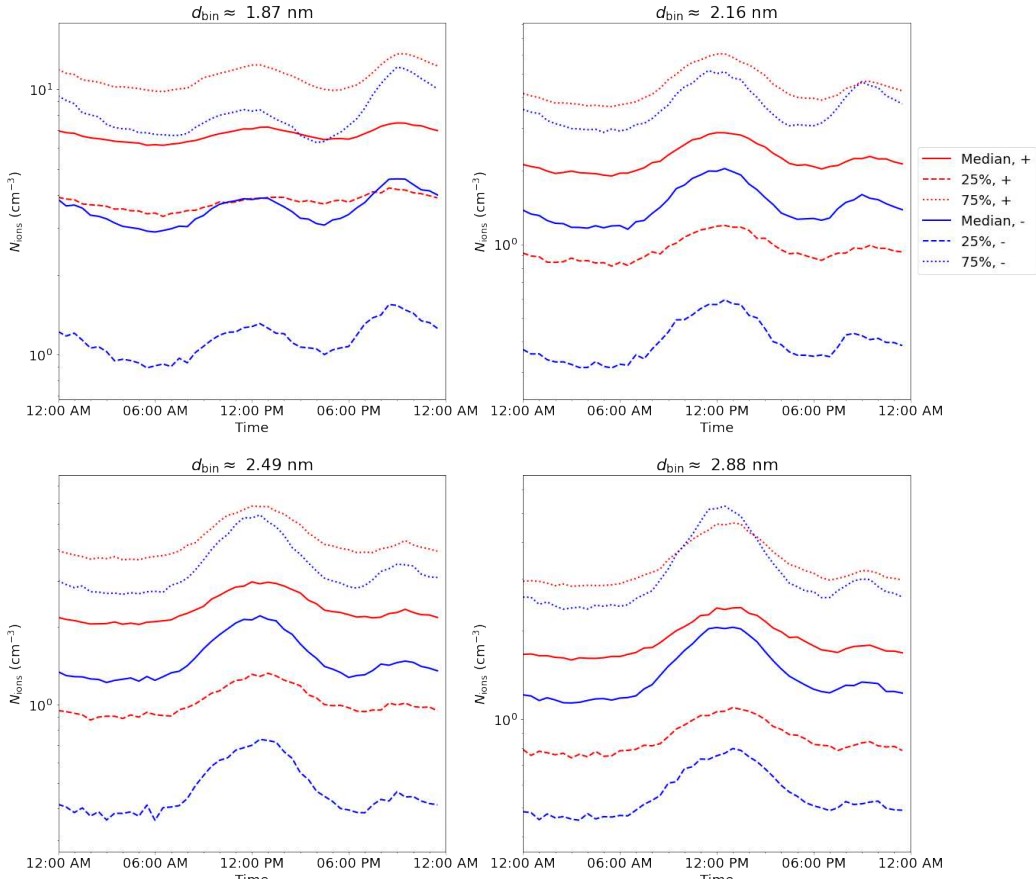

**Figure 1:** Diurnal profiles for ion concentrations ($N_{ion}$) in size bins with median diameter $d_{bin}$ based on median, 25%, and 75% quantiles. The ion concentrations were measured by Neutral cluster and Air Ion Spectrometer (NAIS) at SMEAR II measurement station in Hyytiälä, Finland from 2016 to 2020. Data from all seasons is included and no distinction between the days that were classified as NPF events days or not was made.

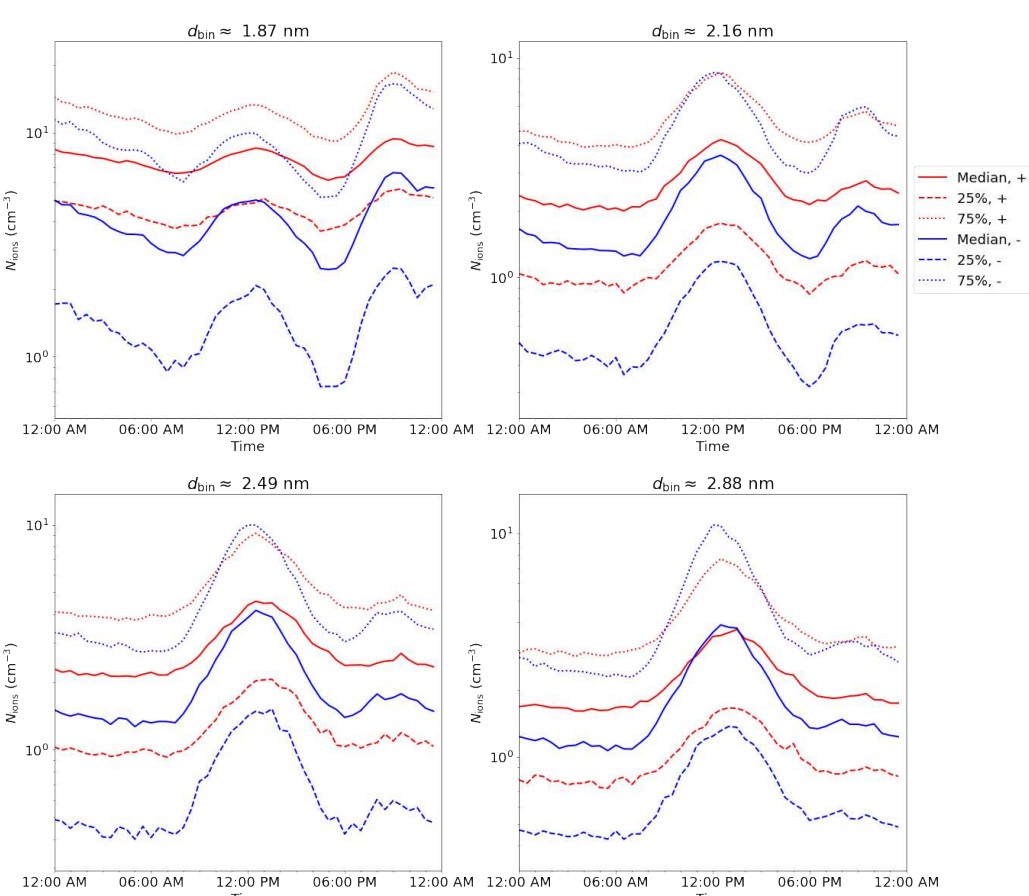

**Figure 2:** Diurnal profiles between March and May for ion concentrations ($N_{ion}$) in size bins with median diameter $d_{bin}$ based on median, 25%, and 75% quantiles. The ion concentrations were measured by Neutral cluster and Air Ion Spectrometer (NAIS) at SMEAR II measurement station in Hyytiälä, Finland from 2016 to 2020.



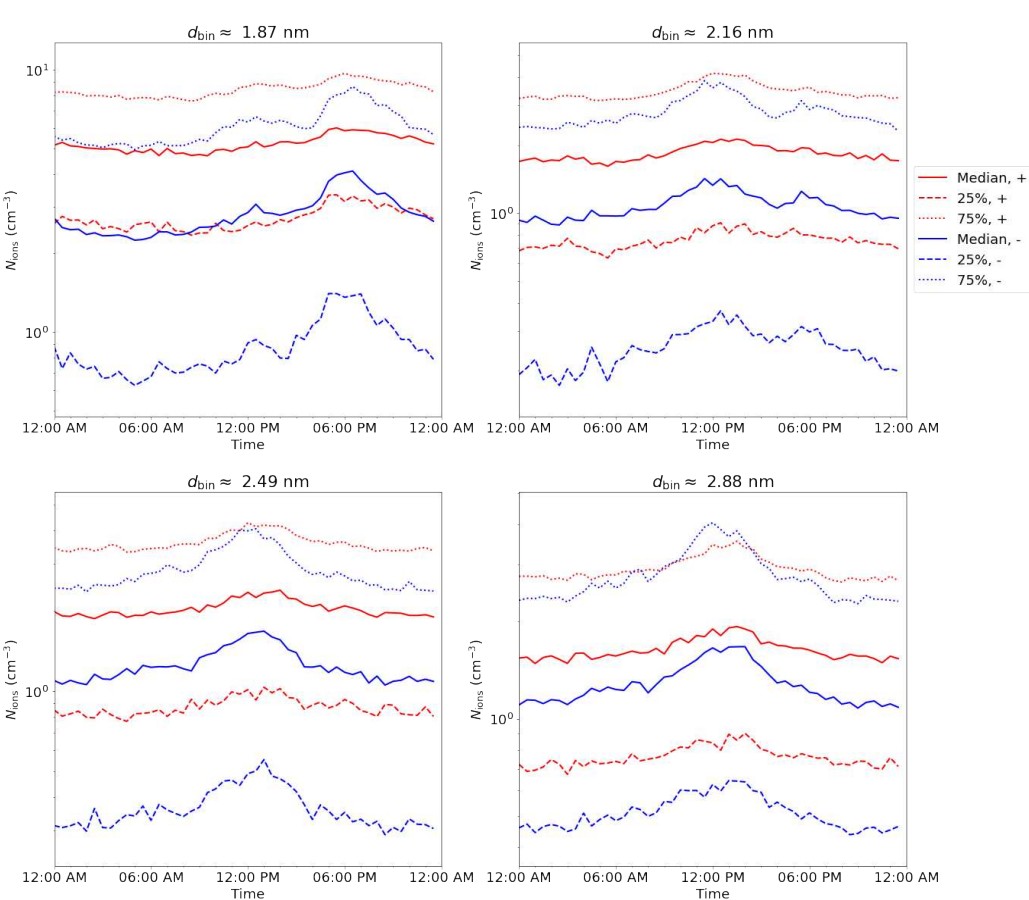

**Figure 3:** Diurnal profiles between September and November for ion concentrations ($N_{ion}$) in size bins with median diameter $d_{bin}$ based on median, 25%, and 75% quantiles. The ion concentrations were measured by Neutral cluster and Air Ion Spectrometer (NAIS) at SMEAR II measurement station in Hyytiälä, Finland from 2016 to 2020.



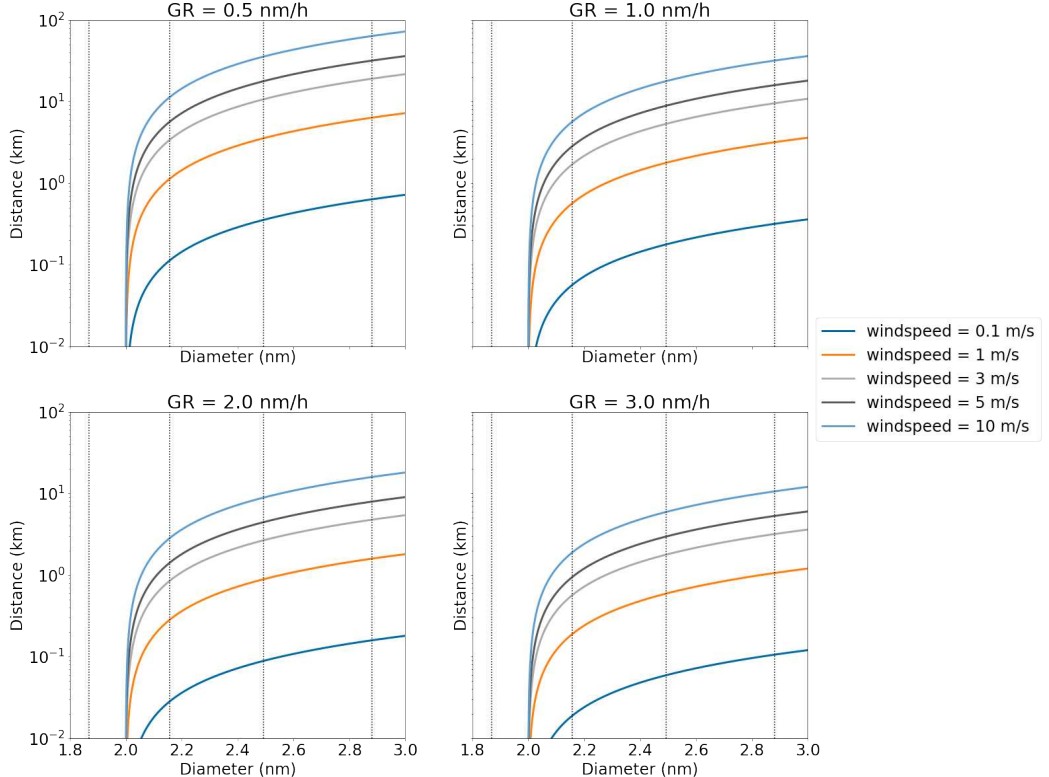

**Figure 4:** The distance a growing atmospheric ion or a neutral particle can be transported by horizontal winds assuming initial diameter of 2 nm. Growth rate of the ion/particle is denoted by GR and it is assumed to stay constant with increasing size.




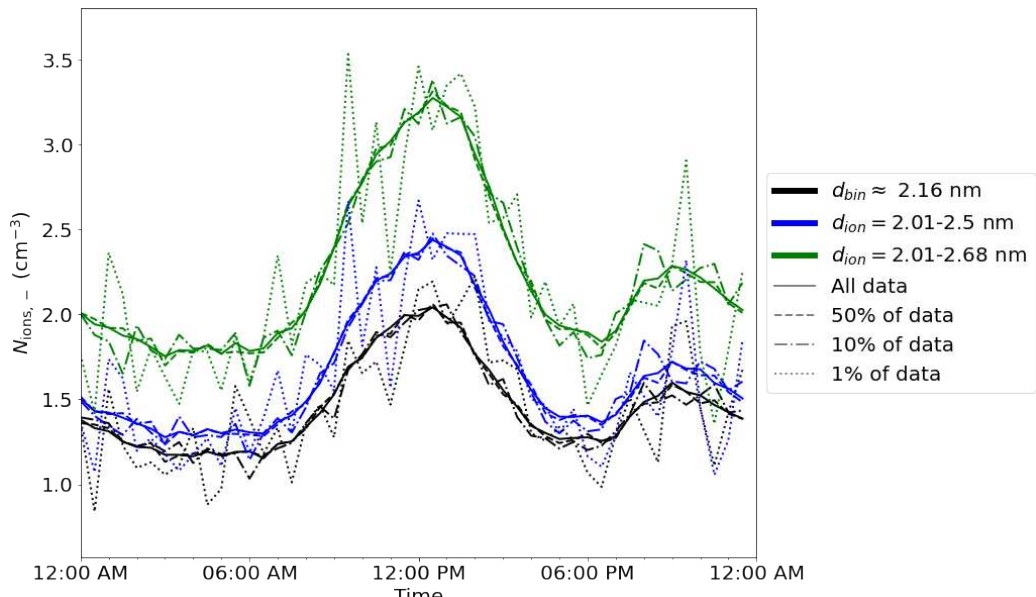

**Figure 5:** Median daily cycle of concentrations of negative ions in $d_{bin} \approx 2.16$ nm and between size limits 2.01-2.50 nm and 2.01-2.68 nm, which include data from both $d_{bin} \approx 2.16$ nm and $d_{bin} \approx 2.49$ nm. Data is from 2016 to 2020, and it was measured with Neutral cluster and Air Ion Spectrometer (NAIS).

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
