# Peer review of "Measurement report: Ion clusters as indicator for local new particle formation"

_EGUsphere, 2023_

## Author Comment (AC1)

**Response to review comments on manuscript egusphere-2023-1108**
*Measurement report: Intermediate ions as indicator for local new particle formation*

**Answers to referee 1:**

The manuscript by Tuovinen et al. presents analysis of Neutral cluster and Air Ion Spectrometer (NAIS) ion data taken at the SMEAR II measurement station in Hyytiälä between 2016 and 2020. From their analysis the authors claim that negative ions between 2.0 and 2.3 nm are best for the characterization of local new particle formation (NPF) events. However, there is a lack of in-depth analysis of the data that supports the claims, an actual applied characterization of a local NPF event is absent and, finally, the writing is missing details and the figure references are incorrect. Therefore, I could not recommend this manuscript for publication in Atmospheric Chemistry and Physics (ACP). Here are the main points that led to the decision.

We thank the referee for the comments, which have been very helpful for us in seeing the weaknesses of our manuscript and on improving them. We sincerely hope that the revised version is better suited for publication. In addition, the first author apologizes for the careless mistake of not noticing that the figure numbers had mixed up, which undoubtedly made it harder to read the manuscript. We have included our answers (in light blue) to the questions (in black) and marked sections from the manuscript with *italics*.

First, we would like to note that, in the revised version of the manuscript, we have chosen to define local formation of intermediate ions (LFII) to refer to the formation of intermediate ions during new particle formation. This change has been done to clarify our aim in this manuscript, which is to use the local formation of intermediate ions to capture and represent NPF. We have changed the title of the manuscript to *Measurement report: intermediate ions as indicator for local new particle formation.* These changes are also reflected in our answers below and in the modifications made to the revised manuscript.

The main analysis is based on the median, 25%, and 75% quantiles of the ion's diurnal cycles (Sec. 3.1). Four different size bins between 1.6 and 3 nm of positive and negative ions were considered. I see several issues with this analysis. The authors just compare seasons where NPF events are observed more regularly (Mar. - May, Fig. 2) to seasons where they are less common (Sep. – Nov., Fig. 3). However, they wrote: "These data were used from all the available days, and no distinction was made based on whether the days had been classified as NPF days or not." I think this is defeating the whole purpose of the analysis. It would be interesting if the ion signal differs during an NPF event independent of the seasons. There is no discussion/analysis of whether the ion signal is influenced by seasons in general, e.g., analyzing only non-NPF days for all the seasons.

We chose to not analyze our data based on the traditional NPF classification due to the recent research on the occurrence of NPF on days, which are usually classified as non-event days (Kulmala et al., 2022). Thus, even if we were to analyze non-events and events separately, we would still have NPF also on the non-event data, it would just be less intense, or the growth of particles to larger sizes would be less effective. As such, we have made the choice to think of NPF as less of an on-off type of phenomenon and more of as a sliding scale with varying levels of intensity from strong regional NPF events to weak NPF hidden in the noise of the measured signal.

We have clarified this in the revised manuscript (Sect. 2.2, lines 94-97):

*Recent advances have shown that NPF does occur even during the days classified as non-event days (Kulmala et al., 2022). As such, we chose to include all the days in the analysis, regardless of whether a NPF event was observed to occur during these days, or not.*

For the actual analysis, the authors only compare peak heights of the median or 75% quantile for the different seasons to conclude that the negative ions in the size bin 2.16 nm are the best for representing NPF. First, the authors do not explain why they use the mainly the 75% quantile for their peak height analysis. Second, the authors did not discuss the spread between the 25% to 75% quantile, which represents the variability of the ion concentration. Third, most of the features of Fig. 2 (Mar. – May) are also visible in Fig. 1 (whole time period). There is no discussion of what this means for distinguishing a NPF signal from a non-NPF signal.

The higher the peak ion concentration value, the more likely there is to be significant NPF during a day, and as such, much of the focus in the analysis has been based on the 75% quantile values. The role of the different quantiles has been clarified in the revised manuscript (Sect. 2.2, lines 104-108):

*The 75% quantile concentrations include, with a high probability, the data that correspond to times of high local formation of intermediate ions (LFII), while the 25% quantile is more likely to include data from times with no LFII. As such, although no strict division between NPF events and non-event was made, we could derive information on the ion concentrations with respect to the strength of LFII, and thus also potential NPF.*

The spread of the concentrations between 25% and 75% quantile have now been briefly addressed in the revised manuscript (Sect. 3.1, 163-167):

*The spread of values between the 25% and 75% quantiles is higher for negative ion concentrations than positive, showing that the 75% quantile concentrations are not as separable from the 25% quantile for positive polarity as they are for negative polarity.*

The argument about excluding the positive ions from further analysis is also very weak in my opinion. In most cases the positive ions have a stronger signal, which might be a plus in this size range below 3 nm, and the spread of the 25% to 75% quantile is most cases smaller. There might be some arguments for the exclusion, but I don't think it is obvious only from the presented data.

We would argue that both the higher signal and the smaller spread of values for positive ion concentrations compared to the negative ion concentrations are a result of the commonly higher background concentrations of the small positive ions compared to the negative ions. As such, these factors can be used in favor of using the negative ion concentrations over positive, in case the former or both are available. Two figures, Figures 1 and 2 below (Fig. 4 and 5 in the revised manuscript) have now been added to the revised manuscript. Fig. 1 shows the 75% quantile concentrations divided by the 25% quantile concentrations, and illustrates the strength of the 75% signal compared to the 25% signal. For all the investigated size bins, and for all the hours, the 75% signal is relatively stronger for negative ions. As such, we argue that LFII should be more clearly observable in the negative ion concentrations compared to positive ion concentrations.

Fig. 2 shows the median ion concentrations divided by the background ion concentration, which was defined as the daily median concentration between 00:00-08:00. Fig. 2 illustrates how the negative ion concentrations during the peak values corresponding to times with the strongest LFII

are relatively higher compared to the background than positive ions. Fig.1 and 2 have been discussed in the revised manuscript (Sect. 3.1, lines 176-191):

*Fig. 4 shows the 75% quantile values divided by 25% quantile values, giving further insight into the differences in the concentration signals between the times of strong LFII versus no, or little, LFII. We see that for all the four size bins the difference between 75% and 25% is higher for negative ions compared to positive ions. For example, for the negative concentrations in $d_{bin} \approx 2.16$ nm, the 75% quantile concentrations are approximately 10 times higher than the 25% quantile concentrations. For the positive ion concentrations in the same size bin, the difference is only by a factor of 5. Thus, the difference between strong LFII and little to no LFII can be expected to be higher for negative ions than positive ions. Fig. 5 shows the median hourly values of the ion concentrations divided by the daily background concentrations (see Sect. 2.2). It shows that, except for $d_{bin} \approx 1.87$ nm, the ion concentrations during the daytime peak, which corresponds to daytime LFII, the negative ion concentrations are higher compared to their background concentration than positive ion concentrations. For $d_{bin} \approx 2.88$ nm, the peak concentration is around 1.7 times higher than the background for negative ions, and around 1.4 times higher for positive. Thus, based on these observations, we can expect LFII to be more easily observable in the negative ion concentrations. However, it is important to note that this only applies to averaged behavior, and individual days might show very different behavior.*

In addition, more discussion in general on the differences between the positive and negative ion concentrations has been now included in the revised manuscript (Sect. 3.1, lines 158-174):

*Most of the ion concentrations in Fig. 1, Fig. 2, and Fig. S2 have an observable peak during midday, indicating the occurrence of daytime LFII. These peaks are only absent from the 25% concentrations in $d_{bin} \approx 1.87$ nm. First, we will discuss the differences between the two polarities. From Fig. 1 the first obvious difference between the concentrations in the two polarities is that the positive ion concentrations appear to be higher compared to the negative ion concentrations, which holds true for all hours. The spread of values between the 25% and 75% quantiles is higher for negative ion concentrations than positive, showing that the 75% quantile concentrations are not as separable from the 25% quantile for positive polarity as they are for negative polarity. In addition, the difference between the peak concentrations and the lower concentrations appears to be higher for negative ions compared to positive ions. If we only look at the 75% quantile concentrations during spring (Fig. 2), which is the season with the most abundant NPF, we can see that, aside from $d_{bin} \approx 1.87$ nm, the peak concentrations for negative ions are equal, or even higher, than for positive ions. This shows that even though the positive ion concentrations on average are higher than the negative ion concentrations, during strong LFII this is not true for $d_{bin} \approx 2.16, 2.49,$ and 2.88 nm. In addition, as the average concentrations for negative ions are lower, the difference between the times when there is strong LFII and when there is little LFII is on average higher for the negative polarity compared to the positive.*

To my understanding, the transport model of Sec. 3.2 consists only of very simple linear equations, which are not mentioned in the text but are easy to derive (distance = wind speed * time; particle size = GR * time + initial size; substitute time; the outcome matches with the presented data). However, the equations and a table would be more illustrative than plotting linear equations in a y-log plot. The initial conditions, taken from other publications, mainly define the outcome, e.g., the initial size favors the choices to the next larger size bin and excludes the first size bin. I think this model is too simplistic for a main finding (defining "local" NPF) in an ACP publication. The authors did not even try to validate the model by any other means.

How these calculations were made has now been clarified in the revised manuscript (Sect 2.3, lines 115-120):

*Simple linear calculations were made to illustrate how long a distance a growing ion can be transported before being measured depends on its size. We assumed a constant growth rate (GR) for the ions, and that the growing ions were transported horizontally along air masses characterized by a constant wind speed. Thus, if the initial ion size is $d_0$ and the size it is measured at is $d_1$, we can say that the farthest distance it can have traveled during its growth is*

$$distance = \frac{d_1 - d_0}{GR} \times windspeed. \qquad (1)$$

The point of these calculations was to simply illustrate how the distance that a growing ion has traveled before being detected depends strongly on its diameter, and that such travel distances could differ considerably even in the sub-3 nm size range of ions, as was the case in this study. An ion with a larger diameter will always have spent more time growing compared to a smaller ion. Therefore, the distance it has traveled compared to a smaller ion before being detected will, in most conditions, be longer. We believe that for our purposes a more accurate and true-to-atmosphere estimate of the contribution of transport on ion concentrations is not needed.

In the revised manuscript, this is addressed in (Sect. 3.2, lines 241-248):

*After the formation of an ion or neutral particle from atmospheric clustering, it can start to grow to larger sizes, and during this growth, which does happen instantly, it can move with the air masses. As such, the ions that we measure do not represent those ions that have formed exactly at or over the site. The larger the ion, the longer is the time it has been growing since its initial formation into a stable growing ion. Consequently, the farther it has typically been transported from where it was formed. We have illustrated this point by very simple linear calculations (see Sect. 2.3) shown in Fig. 6.*

In Sec. 3.3 (Impact of data amount on ion diurnal cycle) a lot of details are missing. The authors do not explain how they reduced the data to 50%, 10% and 1%. For the reduction, did they include all days whether they were NPF events or not? How does this affect the data quality? If the ion size bin should be a good indicator for local NPF events, why not show it on a day where there is local NPF event? Is the ion signal only a result of the statistical analysis or could it be really observed during one NPF event? These and many more unanswered questions arise. In addition, two "Figure Error: Reference source not found" show a certain carelessness, which could be noticed throughout the manuscript.

The missing detail on the sampling for the reduced data has been added to the revised manuscript (Sect. 3.3, lines 286-287):

*The 50%, 10%, and 1% samples of the full data were based on a random sampling of data from all data points.*

The results of this study are mainly applicable for a statistical analysis and longer time series of data. Individual days, and the features and behavior of ion concentrations on individual days are complex. In addition, the uncertainties and noise in the concentrations of small ions during individual days are considerable, and these factors limit the applicability of our results in the analysis of individual days. This has been now briefly addressed in the revised manuscript (Sect.3, lines 136-140):

*It should be noted that our results are mainly concerned with statistical features of atmospheric ion concentrations, made from a relatively large number of observational data. Features of atmospheric LFII on individual days, and how that is observable from ion concentrations, might differ from the statistical observations made in this study due, for example, to, variations in particle formation mechanisms/pathways and meteorological conditions.*

In general, I see the analysis and model as too simplistic to be published in ACP. I would encourage the authors to see their analysis as a starting point to explore the NAIS ion signal in more depth for (local) NPF events and attach this analysis as supporting information to the resulting publication.

**Citation**: https://doi.org/10.5194/egusphere-2023-1108-RC1

[Figure]

**Fig. 1:** Hourly 75% quantile ion concentrations divided by the 25% quantile concentrations of the same hour. The different size bins are denoted with $d_{bin}$. The ion concentrations were measured by Neutral cluster and Air Ion Spectrometer (NAIS) at SMEAR II measurement station in Hyytiälä, Finland from 2016 to 2020.

[Figure]

**Fig. 2:** The median hourly ion concentrations normalized by the background ion concentration. The different size bins are denoted with $d_{bin}$. The ion concentrations were measured by Neutral cluster and Air Ion Spectrometer (NAIS) at SMEAR II measurement station in Hyytiälä, Finland from 2016 to 2020.

**Answers to referee 2:**

Tuovinen et al. utilized a Neutral cluster and Air Ion Spectrometer (NAIS) at the SMEAR II measurement station in Hyytiälä, Finland, to find the best ion size to represent the local new particle formation (NPF). They found that negative ions between 2.0 and 2.3 nm are the best range to represent local NPF. The study can help improve our knowledge of new particle formation. However, this study is not well designed. The analysis is not profound and lacks the necessary statistical analysis to support their conclusion. Moreover, the paper is poorly written, the missing details and figure numbers are incorrect, and some figures are lost. This stud is also not within Atmospheric Chemistry and Physics (ACP) scope. Thus, I suggest rejecting this paper. Please see my comment below to support my decision.

We thank the referee for the comments. The first author apologizes for any confusion caused by the mistake of not noticing that the figure numbering was broken. Large fractions of the result section has now been rewritten, and we hope that the revised manuscript is better written and easier to understand and follow. In addition, two more figures have been added to the results section (Fig. 4 and 5 in the revised manuscript) to deepen our analysis. We have included our answers (in light blue) to the questions (in black) and marked sections from the manuscript with *italics*.

First, we would like to note that, in the revised version of the manuscript, we have chosen to define local formation of intermediate ions (LFII) to refer to the formation of intermediate ions during new particle formation. This change has been done to clarify our aim in this manuscript, which is to use the local formation of intermediate ions to capture and represent NPF. We have changed the title of the manuscript to *Measurement report: intermediate ions as indicator for local new particle*

*formation*. These changes are also reflected in our answers below and in the modifications made to the revised manuscript.

1. In the Methods section, the authors did not include all the necessary details:
    1. It is not clear how did they restrict their analysis to atmospheric ions

The Neutral cluster and Air Ion Spectrometer (NAIS) is able to measure charged particles (ions) separately from total particles. Thus, the concentrations used in the analysis only included the ion concentrations.

    2. It is not clear why did you choose 1.6 and 3 nm. The authors should explain this in the Method section. They explained that ions larger than 3 nm might not be local ions. I do not understand this very well. Do you mean you do not have any ion larger than 3 nm generated at the local? How do you know that? How can you prove this?

In this study, we are interested in atmospheric ions within the context of NPF occuring within a limited spatial scalee, which means that we are interested in ions that have formed within a close proximity to the site, whether they have formed as ions (e.g., through ion induced nucleation) or as neutral particles (through neutral nucleation pathways), which have then become charged. Assuming a GR = 1 nm/h, we know that a 3 nm particle (charged or neutral) has been growing for at least one hour since the formation of a stable growing cluster and during this time it could have traveled a considerable distance before it is detected. If a neutral particle of this size is charged in the proximity of the measurement site, and then detected, the ion can be considered created locally, but the particle itself may have formed farther away. This is addressed in the revised manuscript (Sect. 3, lines 123-135):

*We investigated atmospheric ion concentrations for different diameters to determine the most suitable size, or size range, for representing local intermediate ion formation (LFII), which can be used to capture local NPF. Ions in the sizes of a few nm have been previously used to investigate NPF (see e.g., Kulmala et al., 2013b). In this work, we narrowed the investigated diameters to between 1.6 and 3 nm. These limits were chosen based on our motivations in this study: first, we wanted the source area of the ion formation to be as small as possible. Thus, ions larger than 3 nm in diameter were not considered as they could originate from relatively large distances from the measurement site. Second, we wanted to observe clusters that were already growing to larger sizes, i.e., clusters initiating a measurable NPF. Therefore, the smallest clusters below 1.6 nm were not included in our analysis. Such clusters tend to be present practically all the time, and the dynamics explaining the variations in their concentrations is different than for intermediate ions (e.g., Hõrrak et al., 2000; Hirsikko et al., 2011). As such, increases in the concentrations of these cluster ions do not guarantee the formation of larger particles associated with atmospheric NPF.*

    3. How do you define the background?

Background concentrations refer to the semi-constant concentrations of atmospheric ions not attributable to LFII. The revised manuscript has been rewritten so that the somewhat confusing references to background concentrations have been removed. Now, the background has been defined as the median concentration between 00:00 and 08:00, and it has been determined on a daily basis (Sect. 2.2, lines 109-113):

*In addition, we used the daily background ion concentrations, which were assumed to correspond to the concentrations when no, or little, LFII was taking place. These concentrations were determined as median values between 00:00 and 08:00. This time span was chosen based on a visual analysis of the data.*

These background concentrations have been used to produce Fig. 2 (Fig. 5 in the revised manuscript), which shows the hourly median values of the ion concentrations divided by the daily background concentrations.

4. You should add statistical analysis in this section and show results later.

As the statistical analysis is the basis of the results in this study, we have not made major modifications to the structure of the revised manuscript.

5. Did you use any chemical transport models for section 3.2? How do you calculate GR? Did you include chemical reactions? What is your gas precursors and what is the NPF mechanism? This part is not clear at all. You need to add a description of the model in the method section.

In this section, we made simple linear calculations to illustrate how the larger ions, which have had more time to grow to larger sizes, can have traveled longer distances before being detected. We show that these differences can be considerable, even for the size bins investigated, which are mainly in the sub-3 nm size range. This part of the manuscript has now been clarified by adding the following part to the revised manuscript (Sect 2.3, lines 115-120):

*Simple linear calculations were made to illustrate how long a a growing ion can be transported before being measured depends on its size. We assumed a constant growth rate (GR) for the ions, and that the growing ions were transported horizontally along air masses characterized by a constant wind speed. Thus, if the initial ion size is $d_0$ and the size it is measured at is $d_1$, we can say that the farthest distance it can have traveled during its growth is*

$$distance = \frac{d_1 - d_0}{GR} \times windspeed.$$

We hope that this part of the revised manuscript is now easier to understand.

6. Do you have any meteorological data? Moreover, do you have any gas phase measurements?

Meteorological data or gas phase measurements were not included in the analysis. We do not believe including meteorological data or gas phase measurement in our analysis is necessary for the purposes and the main results of this study.

2. Your results do not convince me.
   1. Why do you only show March, May, September, and November data?

The figures showing data for winter (December to February) and summer (June to August) have now been added to the Supplementary Material of the revised manuscript (Fig. S1 and S2, correspondingly).

   2. Your figure numbers are wrong, so I had difficulty referring to the figures. Also, you did not include the figures you discussed in section 3.3.

The first author apologizes for the confusion. The figure numbering has now been fixed in the revised manuscript. The figure, which was discussed in Sect. 3.3, is Fig. 7 of the revised

manuscript. Unfortunately, the reference to it was broken in the previous version, although it was included in the manuscript.

3. Your figure 1-4 should be violin plots, not line plots. Violin plots can help audiences visualize the distribution and variation.

We believe that line plots with different quantiles are more illustrative for our purposes, as they show more clearly the variation of the concentrations with regard to hour of the day compared to e.g., violin plots or box plots.

4. I do not understand how you could use size to indicate new particle formation. I understand that the ion concentration threshold in a size bin should be a good indicator for local NPFm (e.g., NPF happens when the ion concentration is higher than a value).

We are not proposing that the size itself is an indicator for LFII but the concentrations of ions of that certain size are. We have tried to clarify this in the revised version by being more exact with the wording.

**Citation**: https://doi.org/10.5194/egusphere-2023-1108-RC2

**References**

Kulmala et al. (2022) "Quiet New Particle Formation in the Atmosphere." *Frontiers in Environmental Science* 10 (2022): 912385. https://doi.org/10.3389/fenvs.2022.912385